# GRASP para el problema de fijación de precios basado en preferencias

**Raúl Fauste-Jiménez**
Dpto. de Informática y Estadística
Universidad Rey Juan Carlos
Móstoles, España
r.fauste.2020@alumnos.urjc.es

**Sergio Salazar**
Dpto. de Informática y Estadística
Universidad Rey Juan Carlos
Móstoles, España
sergio.salazar@urjc.es

**Isaac Lozano-Osorio**
Dpto. de Informática y Estadística
Universidad Rey Juan Carlos
Móstoles, España
isaac.lozano@urjc.es

**Jesús Sánchez-Oro**
Dpto. de Informática y Estadística
Universidad Rey Juan Carlos
Móstoles, España
jesus.sanchezoro@urjc.es

## Abstract

Uno de los problemas habituales que sufre todo tipo de comercio es establecer un precio a un producto, un aspecto clave que puede aumentar o disminuir las ventas. En este trabajo, abordamos el problema de fijación de precios basado en preferencias, donde, dado un conjunto de usuarios con sus preferencias de compra y presupuesto que disponen, el objetivo es establecer los precios de los productos para maximizar el beneficio de la compañía. Cabe destacar que cada usuario siempre comprará el producto de mayor preferencia que pueda permitirse, considerando que los productos son ilimitados. La literatura ha demostrado que este problema es $\mathcal{NP}$-Completo donde los modelos matemáticos actuales no se comportan bien con instancias grandes. Recientemente, se ha propuesto un enfoque metaheurístico basado en la metodología *Variable Neighborhood Search* que se sitúa como el estado del arte. En este artículo, presentamos una metaheurística usando la metodología *Greedy Randomized Adaptive Search Procedure*. Adicionalmente, uno de los principales inconvenientes es el reducido número de instancias disponibles. Por este motivo se ha generado un nuevo conjunto de datos y se han evaluado todos los métodos bajo el mismo entorno computacional. Los resultados obtenidos demuestran que nuestra propuesta supera el estado del arte y se sitúa a un 2.85 % del modelo exacto en aquellas instancias donde este logra encontrar la solución óptima.

## 1. Introducción

En los últimos años, el rápido crecimiento del comercio electrónico ha propiciado que el problema de fijación de precios de los productos adquiera una importancia crucial por su directa repercusión en el beneficio de las empresas. De esta motivación, surge la definición de los problemas de tarificación.

Este trabajo aborda el *RPP* (del inglés *Rank Pricing Problem*), un problema de optimización que tiene como objetivo fijar los precios de los productos de una compañia para maximizar su beneficio. El *RPP* fue definido por primera vez en [15] donde se demostró ser un problema $\mathcal{NP}$-Completo. Dicho artículo se centra en una variación del *RPP*, se añade una restricción en la escala de precios por la cual dos productos no pueden tener el mismo precio, algo que es de gran utilidad en industrias donde los productos con varias versiones de distinta gama son el modelo de negocio.

XVI Congreso Español de Metaheurísticas, Algoritmos Evolutivos y Bioinspirados (maeb 2025).

En el *RPP* clásico, se conoce, como información del problema, el interés de los clientes en cada producto, de los cuales comprarán a lo sumo uno de ellos. Estos tienen un presupuesto fijo y clasifican los productos en un ranking del más al menos preferido. Una vez fijados los precios, cada cliente compra su producto preferido de entre los que se puede permitir. Se asume una cantidad ilimitada de cada producto. Sin embargo, existen variaciones del problema donde la cantidad de productos es limitada, *Capacitated Rank Pricing Problem (CRPP)* [3], o donde un cliente puede tener varios productos con el mismo nivel de preferencia *Rank Pricing Problem with Ties (RPPT)* [3].

La variante clásica del *RPP* y que se estudia en este trabajo, se asocia con industrias donde el gasto por parte del cliente es considerable (automóviles, teléfonos móviles, electrodomésticos, ...) y cada cliente pretende, por norma general, no comprar más de una unidad. Precisamente, fue en la empresa *General Motors*, [15] donde se estudió este problema por primera vez. La empresa utilizaba una página web para recomendar vehículos a sus clientes. Para ello, los clientes escribían el presupuesto con el que contaban y ordenaban las características de los vehículos de acuerdo a la importancia que les otorgaban y sus preferencias. Una vez hecho esto, la web mostraba una lista de vehículos ordenados según sus preferencias y ajustados a su presupuesto.

Estudios posteriores se han centrado tanto en la complejidad del problema como en el desarrollo de métodos no exactos para su resolución, es decir, buscar soluciones de alta calidad que no siempre certifiquen la optimalidad.

En [1] se propone un algoritmo de ramificación y poda, para la resolución del *RPP*. En este artículo, se modela el *RPP* con un problema de optimización uni-nivel, basándose en la prueba de [15], donde se demuestra que una solución óptima cumple que los precios de los productos pertenecen al conjunto de los presupuestos de los clientes. Posteriormente, realizan una relajación lineal. Para concluir con su propuesta, antes de ejecutar su función de ramificación y poda, ejecutan un preproceso, que consiste en aplicar a la instancia inicial la función $u$. Esta función asigna a los clientes con mayor presupuesto su producto preferido, al resto de clientes su producto preferido dentro de los que no han sido asignados ya a otro cliente con mayor presupuesto. Esto permite partir de una propuesta de solución a la hora de comenzar a evaluar con el algoritmo de ramificación y poda.

Recientemente en [8] se ha propuesto el uso de una metaheurística para la resolución del *RPP*. Su propuesta consiste en un *Variable Neighborhood Search (VNS)* [6, 13], que se vuelve a aprovechar de la prueba descrita en [15] por la que los precios, han de pertenecer al conjunto de presupuestos. Este algoritmo necesita tres parámetros, $L_0, Q, T$, que se utilizan en la fase de exploración, para generar vectores de precios aleatorios sobre los que partir, concretamente $L_0$ vectores. Una vez definidos estos puntos de partida, se eligen los $Q$ vectores con el valor de la mejor función objetivo y de esos $Q$, se hace una selección de $T$ vectores aleatorios de los pertenecientes a sus vecindarios. En caso de no mejorar el valor de la función objetivo encontrado hasta el momento, se pasa a la fase de perturbación, donde se expande el radio del vecindario y se repite el proceso. Esto permite considerar opciones totalmente distintas, puesto que un pequeño cambio en el vector de precios, puede significar un cambio importante en el valor de la función objetivo.

Este trabajo presenta una aproximación metaheurística basada en el procedimiento de búsqueda adaptativo aleatorio y voraz (*Greedy Randomized Adaptive Search Procedure*) [5] para abordar el *RPP*. En comparación con el estado del arte, la propuesta obtiene mejores resultados en las instancias disponibles, como se analizará en la Sección 4.

El resto del artículo se organiza como sigue. En la Sección 2 se formaliza el problema. El algoritmo propuesto se describe con detalle en la Sección 3. La Sección 4 incluye los resultados obtenidos. Por último, las conclusiones e investigación futura se discuten en la Sección 5.

## 2. Descripción formal

Esta sección presenta una formalización detallada del problema basada en la formulación descrita en el estado del arte [8]. Esta versión del *Rank Princing Problem* asume una demanda unitaria, es decir, cada cliente no comprará más de un producto; presupuesto positivo; y acceso a un suministro ilimitado de productos (es decir, en caso de que un cliente quiera comprar un producto, este siempre podrá hacerlo).

Sea $I = \{1, ..., n\}$ el conjunto de productos ofertados y sea $K = \{1, ..., t\}$ el conjunto de clientes a estudio en la instancia del problema. Cada cliente $k \in K$ tendrá asignado por la instancia del

problema un valor de preferencia para producto $i \in I$ representado por la variable $p_i^k \in \mathbb{R}$. Dados dos productos $i, i' \in I$, la condición $p_i^k > p_{i'}^k$ implica que el cliente $k$ preferirá comprar el producto $i$ al producto $i'$. Por último, el conjunto $B = \{b^1, ..., b^t\}$ denota el presupuesto de los clientes, es decir, el presupuesto $b^k$ representa el precio máximo que puede pagar el cliente $k \in K$.

El *RPP* consiste en identificar el vector de valores reales $S = (s_1, ..., s_n)$ que denota el precio de cada producto $i \in I$. Cada cliente comprará aquel producto que se pueda permitir, es decir, su precio sea menor o igual que su presupuesto ($s_i \leq b^k$) que maximice su preferencia individual. La función objetivo del problema será maximizar el beneficio obtenido por las compras de los clientes.

En la Ecuación (1) se muestra la formulación binivel del problema *RPP*. Para esta formulación se necesitan definir un vector de variables binarias para cada cliente $x^k = (x_1^k, ..., x_n^k)$ que denotarán si el cliente $k$ elige comprar el producto $i$.

$$
\begin{aligned}
&\max_{S} \quad \sum_{k \in K} \sum_{i \in I} s_i x_i^k \\
&\text{s.t.} \\
&\qquad s_i \geq 0, \qquad \forall i \in I,
\end{aligned}
\tag{1}
$$

donde $\forall k \in K$, $x_i^k$ es la solución óptima del problema mostrado en la Ecuación (2).

$$
\begin{aligned}
&\max_{x^k} \quad \sum_{i \in I} p_i^k x_i^k \\
&\text{s.t.} \\
&\qquad \sum_{i \in I} x_i^k \leq 1, \\
&\qquad \sum_{i \in I} s_i x_i^k \leq b^k, \\
&\qquad x_i^k \in \{0, 1\}, \qquad \forall i \in I, \forall k \in K
\end{aligned}
\tag{2}
$$

La Tabla 1 muestra un ejemplo de instancia para el problema *RPP*. En este caso se trabaja con dos clientes $K = \{A, B\}$ y dos productos distintos $I = \{1, 2\}$, aunque el problema asume una numeración de los clientes de 1 a $n$, para el ejemplo se denotan con letras para mejorar su comprensión. La Tabla 1 muestra en la segunda columna, Presupuesto ($b^k$), el presupuesto de cada uno de los clientes y en las columnas 3 y 4 la preferencia de cada cliente por el producto respectivo.

Tabla 1: Ejemplo de instancia del *RPP* con dos clientes y dos productos.

| Cliente ($k$) | Presupuesto ($b^k$) | Productos ($i$) | |
| :---: | :---: | :---: | :---: |
| | | Producto 1 ($p_1$) | Producto 2 ($p_2$) |
| $A$ | 60 | 2 | 1 |
| $B$ | 40 | 2 | 1 |

La solución óptima en esta instancia se obtiene asignando al primer producto el precio de 60 unidades ($s_1 = 60$) y al segundo un precio de 40 unidades ($s_2 = 40$). Con esta asignación el cliente $A$ comprará el primer producto ya que su valor de preferencia es mayor y puede permitirse los dos, aumentando el valor de la función objetivo en 60. Por otro lado, el cliente $B$ no puede permitirse el producto 1, por lo que compra el producto 2 que si puede permitirse, aumentando en 40 unidades el valor de la función objetivo. Por lo tanto, el beneficio obtenido por la solución óptima para esta instancia es de 100 unidades.

En la instancia de ejemplo se muestra cómo la solución óptima se obtiene asignando al vector de precios valores dentro del conjunto de presupuestos de los clientes. En [15] se demuestra que esta es una característica necesaria de la solución óptima del problema, es decir, que el dominio de los valores del vector de precios no es continuo, sino que toma valores en el conjunto de presupuestos. De esta forma, el *RPP* toma la definición de problema de optimización combinatoria, y no continua. Bajo esta premisa [8] realizan una transformación a una formulación uninivel del problema, se redirige a los autores a ese trabajo para una información más detallada del mismo.

# 3. Propuesta algorítmica

La metaheurística *Greedy Randomized Adaptive Search Procedure* (GRASP) se propone por primera vez como solución al problema de cubrimiento de conjuntos en 1989 [4]. Posteriormente, en 1995 adquiere una terminología y estructura de metaheurística de propósito general y se formaliza como tal [5]. Trabajos recientes en problemas de optimización combinatoria han demostrado obtener resultados competitivos [2, 11].

El algoritmo GRASP es un proceso multiarranque donde en cada iteración, se realizan una fase de construcción y una fase de mejora, [5, 14]. La primera fase, la fase de construcción, busca generar una solución diversa para posteriormente mediante un método de intensificación, la fase de mejora, obtener un máximo local. El Algoritmo 1 muestra el esquema completo.

---
**Algoritmo 1** GRASP($\alpha$,I,B)
---
1: $S_b \leftarrow \emptyset$
2: **for** $i \in 1 \dots it$ **do**
3:     $S \leftarrow Constructivo(\alpha, I, B)$
4:     $S \leftarrow BsquedaLocal(S, I)$
5:     **if** $RPP(S) > RPP(S_b)$ **then**
6:         $S_b \leftarrow S$
7:     **end if**
8: **end for**
9: **return** $S_b$

---

El algoritmo, durante un número prefijado de iteraciones (pasos 2-8) aplica sucesivamente las fases de construcción (paso 3) y búsqueda local (paso 4). Por cada solución generada, si es mejor que la mejor solución encontrada hasta el momento, se actualiza (pasos 5-7), devolviendo la mejor de entre todas las soluciones generadas (paso 9).

## 3.1. Fase constructiva

La fase de construcción en GRASP tiene como objetivo generar una solución factible de alta calidad. La característica principal de esta fase de construcción es la inclusión de aleatoriedad en el proceso para favorecer la diversidad entre las soluciones generadas. Para ello, se utiliza un parámetro $\alpha$ que será el responsable de controlar el grado de aleatoriedad del proceso y debe ajustarse de forma empírica. El Algoritmo 2 muestra el pseudocódigo del método constructivo propuesto.

---
**Algoritmo 2** Constructivo($\alpha$,I,B)
---
1: $CL \leftarrow \{(i,b) : \forall i \in I, \forall b \in B\}$
2: $S \leftarrow \emptyset$
3: **while** $CL \neq \emptyset$ **do**
4:     $g_{\text{máx}} = \max_{(i,b) \in CL} g(S, i, b)$
5:     $g_{\text{mín}} = \min_{(i,b) \in CL} g(S, i, b)$
6:     $\mu = g_{\text{máx}} - \alpha \cdot (g_{\text{máx}} - g_{\text{mín}})$
7:     $RCL = \{(i,b) \in CL : g(S, i, b) \geq \mu\}$
8:     $(i,b) \leftarrow RND(RCL)$
9:     $S_i = b$
10:     $CL \leftarrow CL \setminus \{(i, b') : \forall b' \in B\}$
11: **end while**
12: **return** $S$

---

El método comienza creando una lista de candidatos *CL* que se compone de todas las posibles asignaciones de precios del producto $i$ con presupuesto $b$ (paso 1). A continuación, se crea una solución vacía (paso 2) y comienza a iterar hasta completar la solución (pasos 3-11). Cabe destacar que, aunque en la fase de construcción tradicional GRASP el primer elemento se elige de forma aleatoria, en este caso no se produce dicha elección, ya que, de forma experimental, se ha comprobado que se obtienen mejores resultados sin esta elección aleatoria.

En cada iteración, se obtienen el mejor ($g_{\text{máx}}$) y peor ($g_{\text{mín}}$) valor de una función voraz que asigna a cada candidato la preferencia a la hora de ser elegido. En este caso, la función voraz utilizada $g$ es directamente el valor de la función objetivo, pero podría reemplazarse por cualquier otro criterio voraz. A continuación, se calcula un umbral $\mu$ (paso 6) que determinará qué candidatos alcanzan un mínimo de calidad para formar parte de la lista de candidatos restringida *RCL* (paso 7). Este umbral es directamente dependiente de un parámetro $\alpha$ que determinará el grado de aleatoriedad del método. En concreto, si $\alpha = 0$, entonces solo aquellos elementos con un valor de función voraz igual a $g_{\text{máx}}$ podrán entrar en la *RCL*, convirtiéndose en un procedimiento completamente voraz (salvo empates). Por otra parte, si $\alpha = 1$, todos los elementos de la *CL* podrán entrar a la *RCL* ya que se seleccionan todos los candidatos que alcancen el valor de $g_{\text{mín}}$.

Finalmente, el siguiente elemento se elige de manera aleatoria de la *RCL* (paso 8) y se asigna a la solución (paso 9). La lista de candidatos *CL* debe ser entonces actualizada eliminando todas las tuplas de elementos que contienen el candidato elegido $i$ (paso 10). El método devuelve la solución construida $S$ (paso 12).

Una vez explicada y analizada la fase constructiva del algoritmo GRASP, la segunda fase consiste en la búsqueda local descrita en la Sección 3.2.

### 3.2. Fase de mejora

La solución construida en la sección anterior no tiene por qué ser un óptimo local respecto a ninguna vecindad ya que, entre otros aspectos, incluye aleatoriedad en su proceso. Por este motivo, dicha solución se mejora mediante el proceso de búsqueda local descrito en esta sección.

Para definir una búsqueda local es necesario comenzar con el operador de movimiento utilizado. En este trabajo, se propone el uso del operador *Intercambiar*$(S, i, j)$ que intercambiará los precios asignados a los elementos $i, j \in I$. En concreto, dada una solución $S$ compuesta por la asignación:

$$S = (s_1, \ldots, s_{i-1}, s_i, s_{i+1}, \ldots, s_{j-1}, s_j, s_{j+1}, \ldots s_n)$$

el operador *Intercambiar*$(S, i, j)$ intercambiará los valores de los elementos $i, j$, obteniendo la solución:

$$S = (s_1, \ldots, s_{i-1}, s_j, s_{i+1}, \ldots, s_{j-1}, s_i, s_{j+1}, \ldots s_n)$$

Una vez definido el operador de movimiento que se aplicará, es necesario definir la vecindad de soluciones que genera dicho operador. Esta vecindad, denominada $N(S)$, se define como:

$$N(S) = \{S' : S' = Intercambiar(S, i, j) \quad \forall i \neq j \in I\}$$

Dada la vecindad, es necesario definir cómo se realizará la exploración de la misma, para lo que se propone el Algoritmo 3. Cabe destacar que existen dos estrategias de exploración para la búsqueda local. La primera y conocida como *Best Improvement* se queda con la mejor solución del espacio de vecindad de una solución concreta, $N(S)$, y es la aproximación descrita en el pseudocódigo 3. La segunda se conoce como *First Improvement*, modifica la solución en cuanto el algoritmo encuentra una mejor, sin necesidad de explorar todo el vecindario. Su implementación es idéntica al pseudocódigo 3, con la única diferencia de que la búsqueda comienza de nuevo tras encontrar una mejora en el paso 10. Dependiendo del problema de estudio, una aproximación puede proporcionar mejores resultados que otra [7], por lo que se estudian ambas aproximaciones y se compararán los resultados obtenidos.

Analizando el Algoritmo 3, en primer lugar se inicializa la mejor solución, $S_b$ a la solución actual $S$ (paso 1). Después, $\forall i \in I$ se realiza un intercambio con otro producto $j \in I : i \neq j$ (paso 7), se evalúa la nueva solución modificada y si dicha solución es mejor que la mejor solución actual, se actualiza (paso 9). En caso de no mejorar la mejor solución actual, se deshace el intercambio para no continuar a partir de una solución de peor calidad (paso 12). Finalmente, se devuelve la mejor solución encontrada durante la búsqueda, $S$ (paso 16).

## 4. Resultados computacionales

Para realizar los experimentos, se ha utilizado un servidor con un procesador AMD EPYC 7282 (2.8 GHz), 16 cores, 8 GB de memoria RAM, Ubuntu Server 20.04, Java 21 y Gurobi 12.0.

**Algoritmo 3** BúsquedaLocal(S,I)
___
1: $S' \leftarrow S$
2: *mejora* $\leftarrow$ TRUE
3: **while** *mejora* **do**
4:    *mejora* $\leftarrow$ FALSE
5:    **for** $i \in I$ **do**
6:      **for** $j \in I \setminus \{i\}$ **do**
7:        $Intercambiar(S', i, j)$
8:        **if** $RPP(S') > RPP(S)$ **then**
9:          $S \leftarrow S'$
10:          *mejora* $\leftarrow$ TRUE
11:        **end if**
12:        $Intercambiar(S', j, i)$
13:      **end for**
14:    **end for**
15: **end while**
16: **return** $S$
___

En estos experimentos se han utilizado las 3 instancias del trabajo previo[1]. Debido al limitado número de instancias, el conjunto fue ampliado mediante la generación de nuevas instancias de forma aleatoria, considerando como rango de presupuestos $[1, 1000]$, partiendo de un número fijo de: número de clientes $[30, 200]$ y número de productos $[5, 150]$. De esta forma, se generaron 30 nuevas instancias teniendo un total de 33 instancias.

Para poder determinar los mejores valores de los parámetros requeridos por el algoritmo propuesto se ha utilizado *irace* [10], el cual automáticamente encuentra la mejor configuración de parámetros para nuestra propuesta. Los parámetros analizados por irace sobre un conjunto preliminar de un 25 % de instancias han sido los siguientes: como criterio de parada, un número de iteraciones entre $[1, 100]$, el valor $\alpha$ utilizado en el constructivo han sido valores en el rango $[0.00, 1.00]$, y finalmente se ha considerado como otro parámetro cada una de las posibles búsquedas locales planteadas.

Los resultados obtenidos por irace sobre un conjunto preliminar de un 20 % de instancias han mostrado que la mejor configuración consta de 87 iteraciones, $\alpha$ de 0,01 y la búsqueda local basada en *Best Improvement*.

Para poder analizar el rendimiento de nuestro algoritmo, se van a mostrar tres tablas para comparar los resultados obtenidos. La Tabla 2 muestra los resultados para las 3 instancias utilizadas en el algoritmo propuesto en el estado del arte. Seguidamente, la Tabla 3 analizará los resultados obtenidos frente a los valores conocidos por el modelo matemático y, finalmente, se mostrará la Tabla 4 con los mejores resultados obtenidos en las instancias donde el modelo exacto no es capaz de obtener una solución.

Las métricas que se incluyen son: F.O., que indica el valor de la función objetivo; Tiempo (s), que indica el tiempo de ejecución en segundos; GAP ( %), que señala la desviación porcentual de la función objetivo frente al valor óptimo de la instancia y #Óptimo, que muestra si el valor encontrado es óptimo o no. Dichas soluciones óptimas han sido certificadas gracias al modelo matemático implementado.

Tabla 2: Comparativa GRASP frente a VNS.

| | GRASP | | | | VNS | | | |
|---|---|---|---|---|---|---|---|---|
| **Instancias** | **F.O** | **Tiempo (s)** | **GAP ( %)** | **#Óptimo** | **F.O** | **Tiempo (s)** | **GAP ( %)** | **#Óptimo** |
| 30x5 | 807 | 0,18 | 0,00 | 1 | 807 | 600,00 | 0,00 | 1 |
| 30x25 | 1018 | 1,93 | 2,30 | 0 | 960 | 600,00 | 7,87 | 0 |
| 60x50 | 1962 | 39,30 | 2,73 | 0 | 1842 | 600,00 | 8,68 | 0 |
| **Total** | 1262,33 | 13,74 | 1,68 | 1 | 1203,00 | 600,00 | 5,52 | 1 |

___
[1] https://github.com/groupoasys/RPP_VNS_data

Cabe destacar que en [8] no se especifica claramente cuál es el máximo valor de la función objetivo obtenido para cada instancia ni el tiempo de ejecución, ya que únicamente se reporta un informe gráfico. Los resultados que se muestran en la Tabla 2 han sido extraídos tratando de ser lo más precisos posible a partir de las gráficas que aportan en su sección de experimentos. La columna de instancias muestra la cantidad de clientes seguido del total de productos. Cabe destacar la diferencia en tiempo computal de nuestra propuesta frente al VNS (13,74 vs 600,00) con un mejor GAP (1,68 vs 5,52).

La Tabla 3 ilustra los resultados de las instancias en las que Gurobi ha conseguido dar la solución óptima, siendo estas aquellas que cumplen $K \leq 100$ e $I \leq 75$:

Tabla 3: Resultados GRASP con la mejor configuración obtenida con irace: $\alpha = 0.01$ con 87 iteraciones y la búsqueda local *best improvement*.

| | Gurobi | | GRASP | | | |
|---|---|---|---|---|---|---|
| **Instancias** | **F.O** | **Tiempo (s)** | **F.O** | **Tiempo (s)** | **GAP ( %)** | **#Óptimo** |
| 30x5 | 807 | 0,46 | 807 | 0,09 | 0,00 | 1 |
| 30x25 | 1042 | 0,41 | 1018 | 1.93 | 2,30 | 0 |
| 60x50 | 2017 | 6,71 | 1962 | 39,20 | 2,73 | 0 |
| 100x25_0 | 39893 | 9335,90 | 38806 | 28,66 | 2,72 | 0 |
| 100x25_1 | 48137 | 13039,21 | 47203 | 26,15 | 1,94 | 0 |
| 100x25_2 | 47425 | 99261,12 | 46202 | 24,95 | 2,58 | 0 |
| 100x25_3 | 41237 | 30250,49 | 40674 | 25,93 | 1,37 | 0 |
| 100x25_4 | 44474 | 9816,64 | 43379 | 25,92 | 2,46 | 0 |
| 100x50_0 | 43340 | 12454,06 | 41685 | 184,03 | 3,82 | 0 |
| 100x50_1 | 49673 | 9176,31 | 48222 | 159,66 | 2,92 | 0 |
| 100x50_2 | 50678 | 10658,31 | 49404 | 170,55 | 2,51 | 0 |
| 100x50_3 | 49302 | 13558,84 | 46767 | 151,76 | 5,14 | 0 |
| 100x50_4 | 44511 | 13371,61 | 43464 | 173,97 | 2,35 | 0 |
| 100x75_0 | 44462 | 401,09 | 42763 | 576,40 | 3,82 | 0 |
| 100x75_1 | 52178 | 853,59 | 50578 | 480,49 | 3,07 | 0 |
| 100x75_2 | 47816 | 1097,51 | 45933 | 512,38 | 3,94 | 0 |
| 100x75_3 | 46087 | 653,59 | 44083 | 514,62 | 4,35 | 0 |
| 100x75_4 | 45928 | 438,47 | 44442 | 529,95 | 3,24 | 0 |
| **Total** | 38833,72 | 12465,24 | 37632,89 | 201,59 | 2,85 | 1 |

Como se observa en la Tabla 3 el algoritmo GRASP solo encuentra una solución óptima, sin embargo, la desviación media que ofrece $(2,85\ \%)$ ilustra que en la mayoría de instancias se queda cerca de ella. Por otro lado, el tiempo medio de ejecución del modelo matemático es, 12465,24 segundos que, comparado con los 201,59 segundos de la metaheurística propuesta, refleja la baja escalabilidad de Gurobi para la resolución del problema. En términos de promedio de la función objetivo obtenido por el algoritmo propuesto es de 37632,89 frente a 38833,72 obtenido por Gurobi.

Para finalizar, la Tabla 4 muestra los valores de aquellas instancias para las que el modelo matemático no fue capaz de obtener una solución debido a limitaciones de memoria. Como era previsible, el aumento en clientes y productos se ve afectado por el tiempo medio de ejecución en comparación.

Tabla 4: Resultados GRASP en instancias $K \geq 200$ e $I \geq 50$.

| Instancias | F.O | Tiempo (s) | Instancias | F.O | Tiempo (s) | Instancias | F.O | Tiempo (s) |
|---|---|---|---|---|---|---|---|---|
| 200x50_0 | 85146 | 678,62 | 200x100_0 | 90803 | 4929,35 | 200x150_0 | 93015 | 17039,00 |
| 200x50_1 | 85269 | 666,88 | 200x100_1 | 95345 | 4278,20 | 200x150_1 | 96212 | 15124,42 |
| 200x50_2 | 84620 | 675,62 | 200x100_2 | 98065 | 4493,50 | 200x150_2 | 102741 | 15087,95 |
| 200x50_3 | 88054 | 629,00 | 200x100_3 | 94881 | 4338,71 | 200x150_3 | 97056 | 16426,00 |
| 200x50_4 | 88629 | 624,93 | 200x100_4 | 85678 | 4672,00 | 200x150_4 | 99797 | 14056,74 |

En resumen, los experimentos demuestran que el algoritmo propuesto supera al estado del arte, logrando una desviación de un 2,85 % con respecto al método exacto. Además, se presentan resultados para instancias en las que el método exacto no es competitivo, evidenciando la eficiencia y robustez de nuestra propuesta en escenarios más exigentes.

## 5. Conclusión

Este trabajo presenta una solución metaheurística basada en GRASP para el problema *Rank Pricing Problem*. El algoritmo se compone de un método constructivo basado en la función objetivo y una fase de mejora basada en intercambios. Uno de los principales inconvenientes del estado del arte es la limitada cantidad de instancias. En este aspecto, se ha ampliado el conjunto con 30 nuevas instancias de diversos tamaños. Para evaluar los parámetros y las componentes algorítmicas que mejor se adaptan al problema, se ha utilizado irace. Una vez obtenida la mejor configuración, se ha comparado frente a un modelo exacto y al mejor algoritmo conocido en la literatura.

Los resultados muestran cómo frente al estado del arte, nuestra propuesta obtiene mejores resultados en 2 de las 3 instancias. Comparando frente a todas las instancias que resuelve el modelo matemático, nuestro algoritmo se sitúa a un 2,85 de desviación y adicionalmente se han mostrado los resultados para las instancias más grandes.

Dentro de los trabajos futuros de este problema, el principal es obtener el código del estado del arte o realizar una implementación justa para poder evaluar frente al resto de instancias. Con el objetivo de demostrar que estas contribuciones son aplicables a problemas relacionados, nace también el interés en abordar alguna otra variante del problema. Finalmente, también se plantea el uso de otras métricas voraces en la fase constructiva o el uso de métodos de aprendizaje supervisado que contribuyan a la toma de decisiones [9, 12].

## Agradecimientos y declaración de financiación

Los autores agradecen el apoyo del "Ministerio de Ciencia, Innovación y Universidades (MCIN/AEI/10.13039/501100011033/FEDER, UE) bajo la subvención ref. PID2021-126605NB-I00 y PID2021-125709OA-C22, del "Ministerio para la Transformación Digital y de la Función Pública" mediante Concesión TSI-100930-2023-3 y de la Comunidad Autónoma de Madrid, con el proyecto CIRMA-CM (referencia TEC-2024/COM-404).

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
