# OpenReview forum: "GRASP para el problema de fijación de precios basado en preferencias"
_MAEB/2025/Congreso — MAEB 2025_

### Official Review · Reviewer_QZ7a · 2025-03-15
**Propuesta de algoritmo GRASP para el problema de fijación de precios basado en preferencias**

**Rating:** 4
**Confidence:** 4

**Review:**

El artículo presenta un algoritmo GRASP para tratar el problema de la fijación de precios basada en preferencias. Los autores indican que, según sus resultados, su propuesta supera a un algoritmo VNS considerado como el estado del arte para la resolución del problema mencionado con anterioridad. Además, indican que el algoritmo GRASP es capaz de proporcionar soluciones muy cercanas a las óptimas para aquellas instancias donde un método exacto es aplicable.

El trabajo se encuentra bien escrito y organizado y encaja perfectamente con la temática de MAEB. El bajo número de instancias del problema (solo tres en la literatura previa) podría indicar que el problema no se ha trabajado lo suficiente y/o que no tiene aplicación práctica en la realidad. Luego, los autores han generado instancias de manera totalmente aleatoria, lo que hace que las mismas también se encuentren alejadas de lo que podría ser un caso real de aplicación. ¿No sería posible obtener datos proporcionados por alguna empresa para tratar de generar un conjunto de instancias más ajustado a la realidad? También cabría la posibilidad de generar instancias a través de los métodos propuestos por autores como K. Smith-Miles o A. Marrero.

Luego, en la evaluación experimental, se ejecutan algoritmos estocásticos. Para esos algoritmos, ¿cuántas repeticiones de cada ejecución se han llevado a cabo? ¿Las tablas de resultados muestran los valores obtenidos en una única ejecución? Si así fuera, se recomienda repetir cada ejecución de cada algoritmo con cada instancia, con el objetivo de hacer más sólidos los resultados obtenidos. Al mismo tiempo, solo se muestra una comparativa de VNS vs. GRASP para las tres instancias ya propuestas en la literatura. ¿Por qué no aplicaron también VNS para las nuevas instancias generadas aleatoriamente? Me resulta un poco impreciso indicar que GRASP ha superado al algoritmo estado del arte, teniendo en cuenta que solo lo ha hecho en un número tan pequeño de instancias.

Por último, en la introducción se indica que se obtiene un speedup aproximado igual a 30 y que se darán más detalles en la sección de experimentos, lo cual no se ha hecho. En cualquier caso, sería deseable indicar que ese speedup se refiere al tiempo que tarda GRASP vs. modelo ejecutado en Gurobi.

Pros:
- Trabajo bien escrito y organizado.
- Propuesta de un nuevo algoritmo para lidiar con el problema de fijación de precios.

Cons:
- Bajo número de instancias del problema, lo que podría indicar que no tiene aplicación práctica real.
- Evaluación experimental y comparativa de algoritmos poco sólida.

---

### Official Review · Reviewer_kEzk · 2025-03-18
**Propuesta de algoritmo GRASP para resolver el problema de fijación de precios basado en preferencias.**

**Rating:** 4
**Confidence:** 4

**Review:**

Este artículo presenta un algoritmo GRASP para el problema de fijación de precios basado en preferencias. Los resultados obtenidos reflejan que el algoritmo propuesto es competitivo respecto al considerado estado del arte y que constituye una alternativa viable a los algoritmos exactos.

En términos generales, el artículo está bien redactado y presenta el contenido de manera clara y estructurada. Algunas observaciones y sugerencias de mejora:

1. En el abstract, en la primera frase, eliminar la coma después de la palabra “comercio”.

2. En la línea 73 de la Introducción, añadir un punto al final de la frase: “El resto del artículo se organiza como sigue.”

3. En la ecuación (1), corregir la condición s_i >=0, en lugar de s_i <=0.

4. En la ecuación (2), incluir \forall k \in K en todas las restricciones.

5. Corregir la palabra “cómo” en la línea 108 (pag 3) y en la línea 235 (pag 8).

6. En los Algoritmos 1 y 2, los valores que representan las preferencias de los productos para cada cliente, p_i^k, deberían incluirse también como input.

7. En el Algoritmo 2, líneas 4 y 5, se debe definir la función g(S,i,b). Los autores mencionan que g corresponde a la función objetivo. Si se refieren a la función objetivo del problema (2), ¿no dependería también de p_i^k?

8. En el Algoritmo 3, línea 6, corregir “for j \in I \ {i} do”.

9. Los autores deben proporcionar más detalles sobre las tres instancias utilizadas en el trabajo previo. Se entiende (de la Tabla 2) que el número de clientes es 30 y 60, y el número de productos es 5, 25 y 50, respectivamente; sin embargo, sería conveniente que esto se especifique explícitamente. Asimismo, se recomienda incluir una explicación sobre el procedimiento seguido para generar dichas instancias en el trabajo previo. ¿Es similar a la generación de las instancias nuevas?

10. Las tablas 2, 3 y 4 reportan las mejores soluciones obtenidas tras 87 iteraciones del algoritmo GRASP. Sin embargo, sería conveniente considerar un número de ejecuciones del algoritmo por instancia y reportar métricas estadísticas que permitan analizar su comportamiento promedio.

---

### Official Review · Reviewer_47y5 · 2025-03-18
**GRASP para el problema de fijación de precios basado en preferencias**

**Rating:** 3
**Confidence:** 4

**Review:**

This paper is about the Rank Pricing Problem and proposes a GRASP algorithm. The algorithm is very straightforward and I'm mainly focusing on the experimental results section.


The first is that I was wondering why the Gurobi is so good on three instances proposed in [15]. In fact, it is faster than your GRASP except for the smallest one. It may be, that these are instances which are closer to reality than the one you generated.

Was irace used with instances in the training set that are different from the one used in the test set. If not, this should be corrected.

The third is that the Gurobi takes on the larger instances quite some time. There are various reasons for this. The first is that it generates in the time given by the GRASP similar or better solutions than GRASP does, but we do not know yes/no. It would be good, if you can test this. (It is true that the closer one is to the optimal solutions the more time it takes.) The second is that these are more or less random solutions that you generated and the Gurobi results may depend much on the type of instances it sees. Yet, this has to be tested and, in particular, it has to be understood what is the difference between the three instances from another paper (or there is actually none).

---

### Decision · Program_Chairs · 2025-03-20

Accept